# Mid-infrared analogue polaritonic reversed Cherenkov radiation in natural anisotropic crystals

Xiangdong Guo[1,2,7], Chenchen Wu[1,2,7], Shu Zhang[1,2], Debo Hu [1,2], Shunping Zhang [3], Qiao Jiang[4], Xiaokang Dai[1], Yu Duan[1], Xiaoxia Yang [1,2] ✉, Zhipei Sun [5], Shuang Zhang[6], Hongxing Xu [3] & Qing Dai [1,2] ✉

Cherenkov radiation (CR) excited by fast charges can serve as on-chip light sources with a nanoscale footprint and broad frequency range. The reversed CR, which usually occurs in media with the negative refractive index or negative group-velocity dispersion, is highly desired because it can effectively separate the radiated light from fast charges thanks to the obtuse radiation angle. However, reversed CR at the mid-infrared remains challenging due to the significant loss of conventional artificial structures. Here we observe mid-infrared analogue polaritonic reversed CR in a natural van der Waals (vdW) material (i.e., $\alpha$-MoO$_3$), whose hyperbolic phonon polaritons exhibit negative group velocity. Further, the real-space image results of analogue polaritonic reversed CR indicate that the radiation distributions and angles are closely related to the in-plane isofrequency contours of $\alpha$-MoO$_3$, which can be further tuned in the heterostructures based on $\alpha$-MoO$_3$. This work demonstrates that natural vdW heterostructures can be used as a promising platform of reversed CR to design on-chip mid-infrared nano-light sources.

On-chip light sources far below the diffraction limit are largely anticipated for photonic integrated systems[1–4]. Cherenkov radiation (CR) is the electromagnetic radiation emitted by moving charges with a speed faster than light phase velocity inside a dielectric medium, which is a promising light source solution due to its ultrabroad frequency coverage from the microwave to ultraviolet ranges[5,6]. CR is emitted in a cone facing the same forward direction as the fast-moving charges[2,7,8]. To separate the generated light radiation from excited particles, significant efforts have been devoted to using reversed CR (i.e., CR travelling in an opposite direction of the moving charges[2,9]) by designing artificial media with a negative refractive index or negative group-

velocity dispersion (NGVD)[10]. Initially, it was theoretically predicted that reversed CR could be obtained in left-handed metamaterials (LHM) with a negative refractive index (i.e., simultaneously negative permittivity and permeability)[11,12]. Then reversed CR in the microwave band was experimentally demonstrated in artificial LHM[13–15], but the large loss of exquisite artificial structures for optical frequency (i.e., ~10 cm$^{-1}$–10$^6$ cm$^{-1}$) makes it difficult to realize reversed CR at this frequency range[9].

The recent emergence of two-dimensional van der Waals (vdW) materials (e.g., birefringent crystals)[16], which have low-loss hyperbolic phonon polaritons with NGVD in the mid-infrared range, provides a

[1]CAS Key Laboratory of Nanophotonic Materials and Devices, CAS Key Laboratory of Standardization and Measurement for Nanotechnology, CAS Center for Excellence in Nanoscience, National Center for Nanoscience and Technology, Beijing 100190, China. [2]Center of Materials Science and Optoelectronics Engineering, University of Chinese Academy of Sciences, Beijing 100049, China. [3]School of Physics and Technology, Center for Nanoscience and Nanotechnology, and Key Laboratory of Artificial Micro- and Nano-structures of Ministry of Education, Wuhan University, Wuhan 430072, China. [4]College of Physics, Chongqing University, Chongqing 401331, China. [5]Department of Electronics and Nanoengineering and QTF Centre of Excellence, Department of Applied Physics, Aalto University, Espoo 02150, Finland. [6]Department of Physics, University of Hong Kong, Hong Kong 999077, China. [7]These authors contributed equally: Xiangdong Guo, Chenchen Wu. ✉e-mail: yangxx@nanoctr.cn; daiq@nanoctr.cn

competitive platform for reversed CR at the optical frequency[17–20]. Reversed CRs have been theoretically investigated in different vdW materials with NGVD[21]. In particular, it has been shown that phonon polaritons in α-MoO₃ (MoO₃), which have optical frequency response, ultralow-loss, and superior anisotropy[18,22,23], provide an ideal platform for realizing reversed CR at the optical frequency and for achieving multidimensional manipulation of the reversed CR (e.g., in the Reststrahlen band of 820–972 cm⁻¹)[24]. However, reversed CR in MoO₃ has not been observed experimentally due to the large momentum mismatch between the fast-moving charges and phonon polaritons.

Here, we leverage the plasmon supported by metal nanowires (mimic fast-moving charges) to efficiently excite reversed CR with MoO₃ phonon polariton at its type I hyperbolic band. The real-space images of analogue polaritonic reversed CR indicate that changing the direction of moving charges can reshape the radiation distribution of these reversed CRs asymmetrically. Furthermore, the isofrequency contour (IFC) is modulated by stacking the hBN layer on the MoO₃ surface, which increases both the radiation angle and the quality factor of reversed CR. Therefore, our results indicate that the mid-infrared analogue polaritonic reversed CR of vdW materials could open an avenue for designing on-chip nano-light sources.

## Results

### Phonon polaritonic reversed CR in hyperbolic crystals

Like conventional CR, phonon polaritonic CR can be generated when the velocity ($\nu_e$) of the moving charges (i.e., free electrons) exceeds the phase velocity of the phonon polaritons ($\nu_{ph}$, i.e., the speed of light in

the medium). Figure 1a schematically compares the conventional (i.e., forward) and reversed CR. Here, we observe phonon polaritonic reversed CR in a natural hyperbolic crystal, MoO₃. As shown in Fig. 1b, it is a type-I hyperbolic material[22,23] in the frequency range of approximately 958–1010 cm⁻¹ with a positive in-plane dielectric constant $\varepsilon_\perp > 0$ and a negative out-of-plane dielectric constant $\varepsilon_\parallel < 0$, hence exhibiting in-plane NGVD, a necessary condition for achieving phonon polaritonic reversed CR. Figure 1c shows the simulated reversed CR in MoO₃, i.e. electromagnetic field distribution ($Re(E_z)$) excited by fast charges (e.g., $\nu_e = 0.2c$) flying 10 nm above MoO₃, and the wavefronts are marked with black dotted lines (details in Fig. S1 of Supplementary Information). Due to the NGVD of MoO₃, the direction of the phonon polariton energy flow (Poynting's vector $S$) is opposite to the wave vector ($k_{ph}$) which can be derived from the wavefront. As seen, $\theta_{RCR}$, the angle between the moving charges and the radiation energy flow, satisfies $\theta_{RCR} = \theta_k + \theta_S$, where $\theta_k$ is the angle between the wavefront and the moving direction of the charges, and $\theta_S$ is the angle between the wavefront and the radiation energy flow. Based on the relationship of phase difference ($\Delta\Phi$) in triangle ABC, the phase difference of AB ($\Delta\Phi_{AB} = k_e \cdot |\overrightarrow{AB}|$) is equal to the phase difference of CB ($\Delta\Phi_{CB} = k_{ph} \cdot |\overrightarrow{CB}| = k_{ph} \cdot |\overline{AB}| \cdot \sin(\theta_k)$). Therefore, we obtain $\sin(\theta_k) = |k_e|/|k_{ph}|$. In addition, according to $k \cdot v = \omega$, thus

$$\theta_k = \sin^{-1}\left(\frac{|\nu_{ph}|}{|\nu_e|}\right) = \sin^{-1}\left(\frac{|k_e|}{|k_{ph}|}\right) \qquad (1)$$

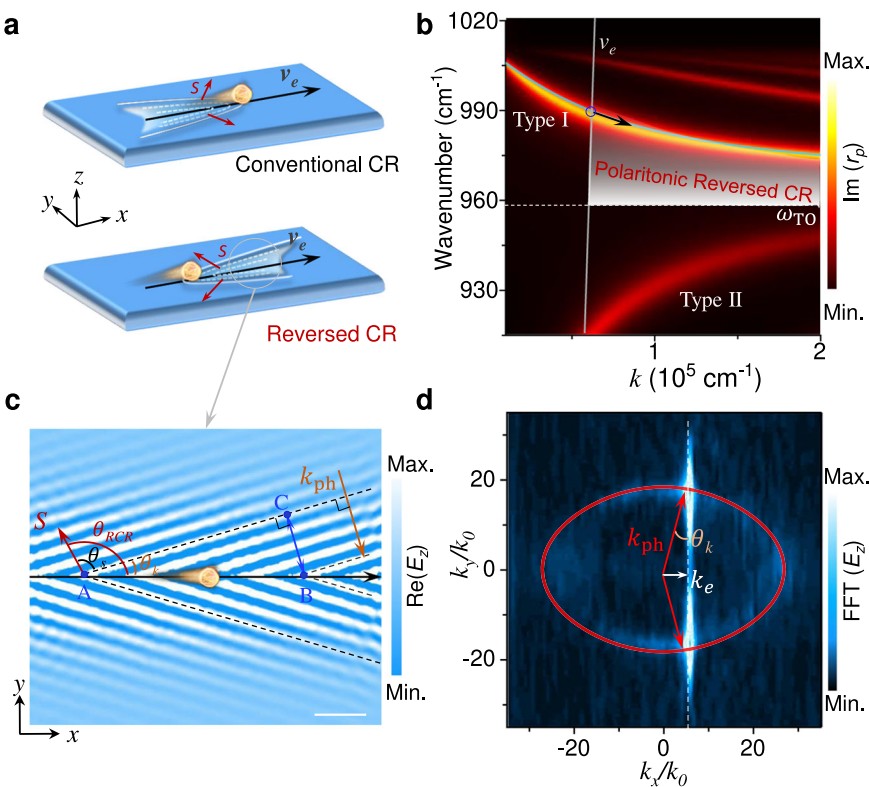

**Fig. 1 | Reversed CR of phonon polaritons. a** Schematic representation of conventional and reversed CR excited via a moving charge. The white lines: direction of the radiated wavefronts. The red arrows: direction of energy flow (e.g, Poynting's vector $S$). The black arrows: movement of electrons with a velocity ($\nu_e$). **b** The anomalous dispersion with negative group velocity to generate reversed CR. The grey line is the dispersion of charged particles at $\nu_e$ ($\nu_e = 0.1c$, c is the vacuum velocity of light). The shaded area represents the frequency range in which the reversed CR is excited at this fixed-charged particle's velocity. **c** Simulated electromagnetic field of the phonon polaritonic reversed CR under moving charged particle excitation ($\nu_e = 0.2c$). The black arrow and black dotted lines represent the moving direction of the charged particles and the radiated electromagnetic waves, respectively. The red and orange arrows represent the direction of $S$ and wave vector ($k_{ph}$) of phonon polaritons. Scale bar: 2 μm. **d** The IFC (red curve) and fast Fourier transform (FFT) of the near field images (in (**c**)). The excitation frequency is 977 cm⁻¹.

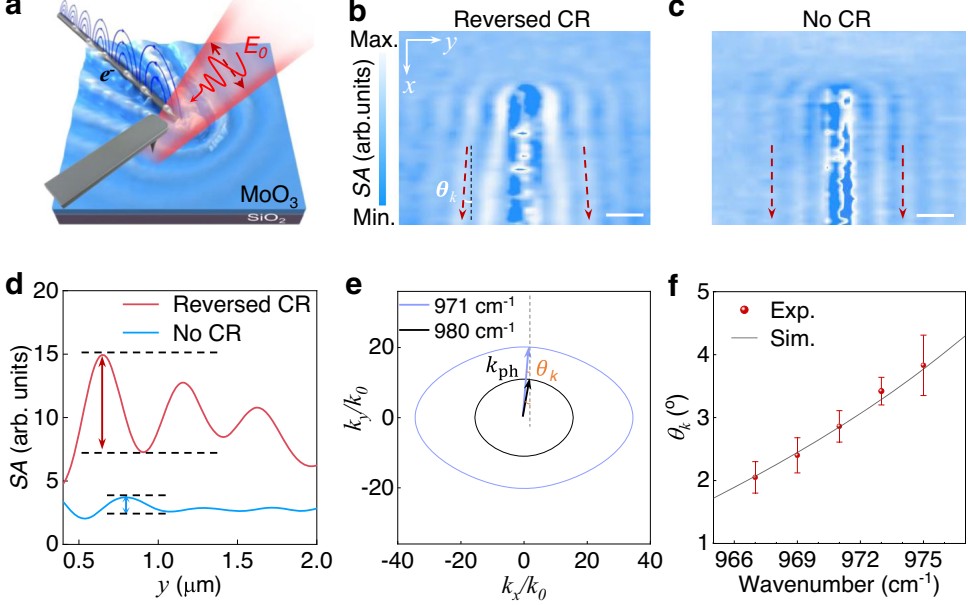

**Fig. 2 | Tunable polaritonic reversed CR. a** Experimental setup for scattering near-field microscopy, where infrared light (polarization field $E_0$) is shining on the metal nanowire in a controlled direction. **b** Experimental MoO$_3$ phonon polaritons with reversed CR (polarization field $E_0$ is along the $x$-axis), and **c**, no CR (polarization field $E_O$ is along the $y$-axis) with an excitation frequency of 971 cm$^{-1}$, respectively. The red dashed arrows are guidelines. The thickness of MoO$_3$ is about 380 nm. $\theta_k$ is equal to the angle between the longitudinal axis of metal nanowire and interference fringes of phonon polariton in MoO$_3$. $SA$ is the near-field amplitude. Scale bars: 0.5 μm. **d** Extracted interference fringes of phonon polaritons at the same position

(1.6 μm from the top of Ag nanowire) from **b**, and **c**. **e** The theoretically calculated IFCs of MoO$_3$ show that the wave vectors of polariton are shrunken as the excitation frequency increases (i.e., increased $\theta_k$). **f** The summarized experimental and simulated extracted $\theta_k$ of MoO$_3$ phonon polariton at different excitation frequencies. Experiment: red points. Simulation: grey line. Each error bar corresponds to a different line profile within a single scan image. Note: In the rectangular coordinate system, the $x$ direction is the direction of the nanowire, and [1 0 0] of MoO$_3$ in Fig. 2 is parallel to the nanowire.

However, $\theta_S$ for the anisotropic MoO$_3$ cannot be quantitatively determined in the real-space, but can only be extracted from the anisotropy IFCs in the reciprocal space.

We perform a fast Fourier transform (FFT) on the near-field mapping obtained in Fig. 1c, which is consistent with the calculated elliptical IFC (red line) in the reciprocal space (see Methods, Fig. 1d and Fig. S2a of Supplementary Information). According to the geometric relationship in reciprocal space, $\theta_S$ satisfies: $\theta_S = \pi/2 - \theta_{S'}$, where $\theta_{S'}$ is the angle between the tangent of the IFC and the wavefront. Thus, $\theta_{RCR} = \theta_k + \theta_S = \pi/2 + \theta_k - \theta_{S'}$ in Fig. S2a and Fig. 1d. Therefore, $\theta_{RCR}$ is greater than 90°, which means that the radiation energy flow is opposite to the direction of moving charged particles (i.e., reversed CR)[3,25]. This analysis in the reciprocal space not only shows the momentum-matching relationship of anisotropic reversed CR, but also provides a probe to the properties of materials.

**Demonstration of phonon polaritonic reversed CR in MoO$_3$**

The CR effect is electromagnetic radiation emitted when a charged particle passes through a dielectric medium at a speed greater than the phase velocity of light in the medium. Due to the challenges of combining a dielectric medium with fast-moving charged particles in a vacuum environment[14,15,26], an optical analog of CR has been developed as an ingenious way to study the physical effects of CR in depth[1,13,25,27,28]. Based on the above theoretical analysis, we exhibit the imaging of in-plane reversed CR in MoO$_3$ in real space via scattering scanning near-field microscopy (s-SNOM), as illustrated in Fig. 2a. Compared with the far-field method of measuring CR[13–15], this near-field method can provide polaritonic reversed CR distribution with approximately 10 nm spatial resolution, such as the radiation wavefront and radiation angle of CR. In order to excite the polaritonic reversed CR, plasmon (i.e., a dynamic charge density wave), which is analogous to a superluminal charge, propagating in a one-dimensional metal nanowire, serves as

the radiation emitter. It is noted that in the mid-infrared, the phase velocity ($\nu_{pe}$) of the plasmon along an Au/Ag nanowire is close to the free-space light velocity ($c$) and is much larger than that of the phonon polaritons[4,29,30].

In the s-SNOM measurement, an obliquely incident infrared light forming an angle of approximately 38° with the sample surface is used to excite the plasmons on the Ag nanowire for visualizing the wavefront of the analogue polaritonic reversed CR. Two typical s-SNOM images of the MoO$_3$ with different polarizations of the incident light are illustrated, which correspond to the excitation of reversed CR (Fig. 2b), and the ordinary phonon polariton without CR (Fig. 2c), respectively. There exist obvious differences in the direction of the measured fringes between two different incident polarizations (red dashed arrows, details in Fig. S2b, c of Supplementary Information). The V-shaped (parallel) bright fringes originate from the constructive interference between the polaritonic CR excited by the plasmon in the metal nanowire and the tip-reflected light, and the observed forward V-shaped wake at 971 cm$^{-1}$ is indicative of a reversed CR (Fig. 2b). By extracting the interference fringes of phonon polaritons for reversed CR and no CR at the same position, we find that the strength of interference fringes with reversed CR is approximately 6 times higher than that of without CR, as indicated in Fig. 2d. This is because the polaritonic reversed CR is excited by the propagating plasmons in the Au/Ag nanowire. In addition, polaritonic reversed CR maintains the coherent performance of phonon polaritons[4,31,32].

To further quantify the analogue polaritonic reversed CR, $\theta_k$ is obtained to determine the direction of the reversed CR from Eq. (1), which depends on $|k_{ph}|$ and $|k_e|$ in the IFCs. It is noted that the wave vector of gold or silver plasmons in the mid-infrared range is approximately equal to free-space light $k_0$ (e.g., $|k_e| \approx |k_0|$, see details in Fig. S3 of Supplementary Information). The IFC shrinks as the excitation frequency increases due to the negative group-velocity dispersion

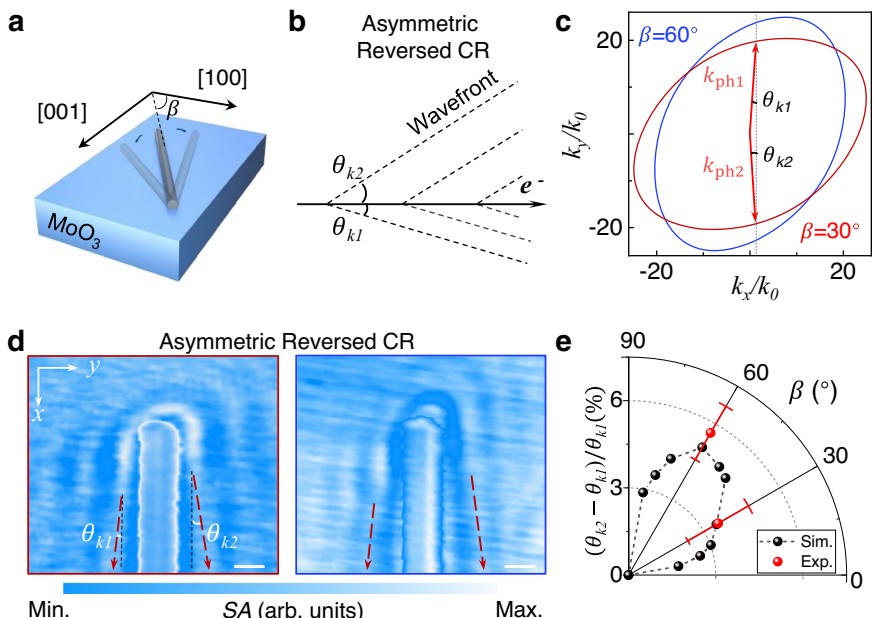

**Fig. 3 | Asymmetric reversed CR in MoO₃. a** Schematic diagram of rotated metal nanowires on MoO₃ with different angles. β is the angle between the metal nano-wire and the [100] direction of MoO₃. **b** Schematic illustration of the MoO₃ phonon polaritons with asymmetric reversed CR. **c** Theoretical prediction of IFCs when β is 30° and 60°. **d** Experimental asymmetric reversed CR of MoO₃ phonon polaritons at 977 cm⁻¹ when β ≈ 30° (left) and 60° (right). Polarization field $E_O$ is along the x-axis. Scale bars: 0.5 μm. **e** The summarization of extracted experimental and simulated asymmetric reversed CR with the value of $(\theta_{k2} - \theta_{k1})/\theta_{k1}$ for MoO₃ phonon polariton at 977 cm⁻¹. Experiment: red points with error bar. Simulation: black points and lines. The thickness of MoO₃ is approximately 280 nm.

of the MoO₃ phonon polaritons, which results in increased $\theta_k$ at higher frequencies, as illustrated in Fig. 2e. We experimentally measure the reversed CR in MoO₃ at different frequencies, with the Cherenkov angle of the wavefront $\theta_k$ summarized in Fig. 2f. These experimental results validate the theoretical expectation that the $\theta_k$ increases at higher frequencies.

**Asymmetric reversed CR in MoO₃**

Directional radiation is important for on-chip CR sources, which facilitates the flexible design of on-chip integrated devices and photon manipulation. The reversed CR of phonon polariton with an elliptical IFC has an inherent advantage for manipulating asymmetric reversed CR directions. Here, we change the angle between the metal nanowire and the [100] of MoO₃ to mimic the direction of the moving electrons for achieving asymmetric reversed CR, which serves as a new degree of freedom to manipulate the reversed CR. For example, when the orientation of the metal nanowire is varied on the MoO₃ as shown in Fig. 3a, the reversed CR angles are generally asymmetric (i.e., $\theta_{k1} \neq \theta_{k2}$) and they change with the orientation angle (β), as shown in Fig. 3b. $\theta_{k1}$ ($\theta_{k2}$) is the reversed CR angle of the wavefront on the right (left) side of the fast charges, which can be calculated by Eq. (1).

The observed asymmetric reversed CR on the MoO₃ with angular dependence can be explained via the elliptical IFC. The elliptical profile with arbitrary rotation angle β can be obtained by rotational trans-formation with a rotation matrix. The predicted IFCs of two typical β values (e.g., 30° and 60°) in the reciprocal space are illustrated in Fig. 3c. It is worth highlighting that the value of $k_{ph1}$ is not the same as the value of $k_{ph2}$ except at β = 0° or 90° due to the in-plane anisotropy of MoO₃. Thus, the asymmetric reversed CR can be excited by plasmons.

As predicted, the asymmetric interference fringes ($\theta_{k1} < \theta_{k2}$) are observed on both sides of the Au nanowire with β of approximately 30° and 60° measured at 977 cm⁻¹ (see Fig. 3d). In addition, the wavefront and propagation distance of the analogue polaritonic reversed CR are both sensitive to the directions of the plasmonic wave vector. This is mainly because $k_{ph}$ is smaller in the direction of [100] crystal axis than

that in the direction of [001] crystal axis for MoO₃. We further quan-titatively calculate the difference $((\theta_{k2} - \theta_{k1})/\theta_{k1})$ of the asymmetric reversed CR, which is strongly dependent on β as shown in Fig. 3e. The asymmetric radiation difference can reach about 3.5% and 5.6% for β≈30° and β≈60°, respectively. To further investigate this phenom-enon, a numerical study on other thicknesses of MoO₃ is performed to demonstrate similar symmetric and asymmetric reversed CRs (details in Fig. S4 of Supplementary Information). This means that the direc-tion of the plasmon oscillation relative to the in-plane crystal axis of MoO₃ provides a new degree of freedom for engineering the reversed CR.

**Reversed CR in hBN/MoO₃ heterostructure**

To further increase the radiation angle ($\theta_k$) and improve the radiation efficiency ($\eta \approx |k_e|/(|k_{ph}| - |k_e|)$)[4] of the analogue polaritonic reversed CR, one can increase the plasmonic wave vector ($k_e$) or reduce the phonon polaritonic wave vector ($k_{ph}$). A higher reversed CR efficiency can be achieved by increasing the coupling between plasmons and phonon polaritons (i.e., decreasing the momentum mismatch between the large $|k_{ph}|$ and low $|k_e|$), which makes it easier for plasmons (source) to transfer energy and momentum to phonon polaritons (the excited reversed CR). The increase of $|k_{ph}|$ and decrease of $|k_e|$ can be realized by increasing the environmental dielectric constant. This is because the plasmon has a positive group-velocity dispersion while the phonon polaritons supported by MoO₃ have a negative group-velocity dispersion[33,34]. Therefore, an effective way to improve the performance of the reversed CR is to construct a heterostructure system by inserting a dielectric film with a high-refractive index between MoO₃ and metal nanowires[35]. Here, we specifically design hBN/MoO₃ heterostructures (Fig. 4a) for this purpose mainly because:[36,37] (1) It has a high refractive index and low optical dielectric loss within the frequency range of approximately 958–1010 cm⁻¹; (2) It has an atomically flat surface to protect MoO₃ and thus reduce the defect losses on its surface.

Figure 4b shows the calculated IFCs of MoO₃ (i.e., air/MoO₃) and hBN/MoO₃ heterostructure. In the hBN/MoO₃ heterostructure, $k_{ph}$ is

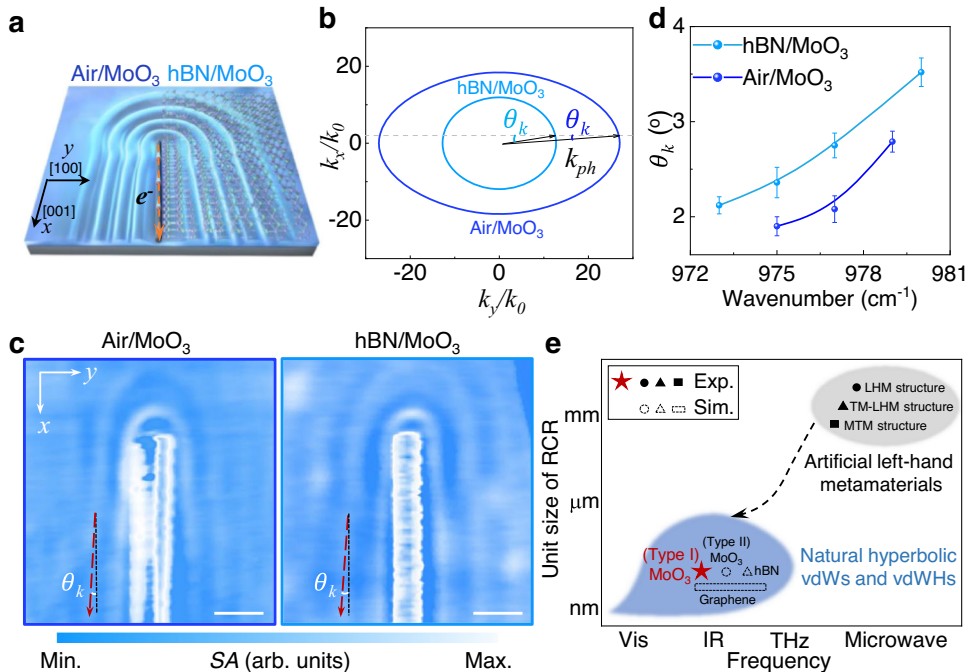

**Fig. 4 | Reversed CR in hBN/MoO₃ heterostructure. a** The schematic illustration of reversed CR in the air/MoO₃ and hBN/MoO₃ heterostructures. **b** The IFC of hBN/MoO₃ heterostructure shrinks compared to pure MoO₃ (i.e., air/MoO₃), so the $\theta_k$ is increased. The thickness of hBN is 50 nm. **c** Real-space images of polaritonic reversed CR in air/MoO₃ and hBN/MoO₃ with an excitation frequency of 977 cm$^{-1}$ when β ≈ 90°. Scale bars: 0.5 μm. **d** The summarized experimental extracted $\theta_k$ on air/MoO₃ and hBN/MoO₃ at different excitation frequencies. Experimental data: points. The thickness of MoO₃ and hBN is approximately 280 nm and 7 nm, respectively. Error bars are obtained from different line profiles within each scanned image. **e** Summary of reversed CR based on varied materials and structures. In the previous experiments, the reversed CR was mainly realized in the microwave band based on the design of artificial left-hand metamaterials (LHM), such as LHM structure[14], transverse magnetic-LHM (TM-LHM) structure[13] and metallic metamaterial (MTM) structure[15]. In the shorter wavelength band, some theoretical works predict that reverse CR can exist in natural hyperbolic vdW materials (vdWs) or vdW heterostructures (vdWHs), such as hBN (type I hyperbolic band)[21], MoO₃ (type II hyperbolic band)[24] and multilayer graphene stacked hyperbolic heterostructures[49].

smaller than that of air/MoO₃, which can increase the angle $\theta_k$ and the radiation efficiency η of analogue polaritonic reversed CR. Moreover, the enhanced electric field is confined in the hBN nanocavity for Au/hBN/MoO₃ structure to improve the excitation intensity of analogue polaritonic reversed CR, as shown in Fig. S5. In the simulation, we calculate the quality (Q) factor for different thicknesses of hBN, and it is found that the Q factor is maximized when the thickness is in the range of 6−9 nm (details in Fig. S5−S7 of Supplementary Information). Therefore, a few-layered hBN (thickness≈ 7 nm) is exfoliated and sandwiched between the Au nanowire and the MoO₃ flake in the experiment (details in Methods). As expected, the performance of reversed CR on the hBN/MoO₃ heterostructure is significantly improved compared to that of MoO₃ (see Fig. 4c, details in Fig. S8−S10 of Supplementary Information). For example, $\theta_k$ observed on the hBN/MoO₃ heterostructure is larger than on the MoO₃. Meanwhile, $\theta_k$ observed on the hBN/MoO₃ heterostructures increases with increasing excitation frequency in Fig. 4d. Furthermore, benefitting from the low loss of phonon polariton, the experimentally observed polaritonic reversed CR exhibits high Q factors and radiation efficiency η, such as Q ≈ 10, η ≈ 3.8% in air/MoO₃ and Q ≈ 16, η ≈ 5.1% in hBN/MoO₃ (details in Fig. S5 of Supplementary Information).

## Discussion

More material systems could be exploited to realize reversed CR in type I hyperbolic band. In Fig. 4e, we summarize the possible implementation of reversed CR based on the NGVD, covering both the artificial structures and the potential natural materials in terms of frequency and unit size of reversed CR. The artificial LHM designed to observe the far-field spectral response of reversed CR is mainly in the microwave band due to the relatively low loss[13–15], while the reversed

CR in natural crystals is determined by the supporting materials. As hyperbolic natural materials have been widely discovered, their type I hyperbolic band can cover from the terahertz frequency range to the visible frequency range[38–41], which indicates the reversed CR can be explored in a wide range beyond the mid-infrared of MoO₃. The experimental strategy can be applied to a large amount of type I hyperbolic materials which have been cataloged in Ref. 42. Moreover, with the strategies of constructing vdW heterostructures, it will be possible to integrate the working frequency ranges of reversed CR in different materials such as the type I hyperbolic bands of hBN and MoO₃ in the heterostructure.

In conclusion, we have demonstrated reversed CR at mid-infrared based on natural hyperbolic materials (e.g, MoO₃). By studying the real-space images of MoO₃ phonon polaritons, it is revealed that the radiational distribution and reversed CR angle can be effectively modulated. This is achieved by changing the IFC and the direction of moving charges relative to the crystal axis. In addition, by constructing the hBN/MoO₃ heterostructure, we can not only tune the radiational angle but also improve the Q factor of reversed CR by approximately 60%. The reversed CRs on vdW heterostructures establish a possible route for on-chip free-electron infrared sources, which reveals strong light-matter interactions and shows potential for long-sought-after applications in nano-optoelectronics.

## Methods

### Nanofabrication of the devices

High-quality MoO₃ flakes and hBN films were mechanically exfoliated from bulk crystal (2D Semiconductor) with the tape (Nitto), then transferred onto commercial 285 nm SiO₂/500 μm Si substrates (SVM). The hBN/MoO₃ heterostructure was prepared by transferring an hBN

film onto a MoO$_3$ flake with the help of a polydimethylsiloxane (PDMS) stamp (Shanghai Onway Technology Co., Ltd). We chose metal (silver or gold) nanowires to excite CR. The silver nanowires were spin-coated on the MoO$_3$ flakes (silver nanowires dispersed in alcohol) synthesized by using a polyol process[43,44]. A series of Gold nanowires (100 nm-width and 10 μm-length) were patterned on selected MoO$_3$ flakes and hBN/MoO$_3$ heterostructure by coating approximately 350 nm poly(-methyl methacrylate) (PMMA) 950 K lithography resist and utilizing 100 kV electron-beam lithography (Vistec 5000 + ES). Then 10 nm Cr and 100 nm Au are evaporated and deposited by Electron-beam in a vacuum chamber at a pressure of $< 1 \times 10^{-6}$ torr. Finally, hot acetone is used to remove PMMA on the samples, and then samples were subjected to gentle washing with isopropyl alcohol (IPA), followed by drying with flowing nitrogen gas.

### Scanning near-field optical microscopy measurements

A scattering scanning near-field optical microscope (Neaspec) with a wavelength-tunable quantum cascade laser was used to image near optical fields. The probes are made initially for metalized atomic force microscope (AFM) with an apex radius of approximately 10–20 nm (Nanoworld), and the tip-tapping frequency was set to approximately 270 kHz. The *p*-polarized IR light from the monochromatic quantum cascade lasers is focused via a parabolic mirror onto both the tip and sample at an angle of approximately 52° relative to the tip axis[45].

### Electromagnetic simulations and theory

The phonon polaritonic reversed CR is calculated by finite-element method (FEM) simulation using COMSOL Multiphysics[46]. With the input of thickness and ω from our experiment, we simulated the real-space electromagnetic field $E_z$ and modelled momentum space (*k*-space) IFC. In the real-space simulation, the polarization and direction of the incident mid-infrared light are the same as in the experiment. The IFC and dispersion can be calculated as a transfer matrix from ref. 22. The permittivity of the MoO$_3$ layer and hBN layer is obtained by fitting the experimental results in refs. 22,23,47,48. with a Lorentzian model.

## Data availability

The data that support the findings of this study are available from the corresponding author upon reasonable request. Source data are provided with this paper.

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

## Acknowledgements
This work was supported by the National Key R&D Program of China (2021YFA1201500), the National Natural Science Foundation of China (51925203, 52022025, 52102160, 51972074, and U2032206), the Strategic Priority Research Program of the Chinese Academy of Sciences (XDB36000000 and XDB30000000), Youth Innovation Promotion Association C.A.S., C.A.S. Interdisciplinary Innovation Team (JCTD-2018-03), the Open Project of Nanjing University (M34034), the Special Research Assistant Program of the Chinese Academy of Sciences, the Young Elite Scientists Sponsorship Program by CAST (2022QNRC001), the Research Grants Council of Hong Kong (AoE/P- 701/20, 17309021), Academy of Finland (314810, 333982, 336144, 352780, 352930, and 353364), the Academy of Finland Flagship Programme (320167,PREIN), the EU H2020-MSCA-RISE-872049 (IPN-Bio), and ERC (834742).

## Author contributions

The concept for the experiment was initially developed by Q.D., X.Y., and X.G. s-SNOM experiments were performed by C.W. and X.G. under the direction of Q.D., D.H., and X.Y.; Metal nanowires and MoO3 samples were prepared by C.W., assisted by Shu Zhang; Ag nanowires were prepared by Shunping Zhang and H.X.; FEM simulations and theoretical analysis were performed by X.G. assisted by Shuang Zhang; Experimental data processing and analysis was performed by C.W. and X.G. assisted by Q.J., X.D., and Y.D.; X.G. and C.W. wrote the manuscript with advice from Shuang Zhang, Z.S., X.Y., and Q.D.; X.G. and C.W. contributed equally to this work. All authors discussed the results at all stages and participated in the development of the manuscript.

## Competing interests
The authors declare no competing interests.
