## [Peer Review File · Nature Communications]

Reviewers' Comments:

Reviewer #1:

Remarks to the Author:

This paper reports a so-called reversed Cherenkov radiation (CR) in the mid-infrared band using natural anisotropic crystals. It is an interesting topic for readers. Here, the comments are listed below.

1. The authors claim that the reversed CR occurs in natural materials. This claim is NOT consistent with the Veselago's viewpoint.
2. CR is the electromagnetic radiation emitted by moving charges with a speed faster than the phase velocity of the electromagnetic wave inside a medium. However, in fact, the actual moving charged particles are not employed in the paper. The so-called reversed CR should be considered as analog of reversed CR or a special case of negative refraction not REAL reversed CR. This is because CR clearly shows the particle-wave interaction NOT wave (here light)-matter interaction. Please refer to the following two papers:
(1) Grbic, A. and Eleftheriades, G. V. Experimental verification of backward-wave radiation from a negative refractive index metamaterial. *Journal of Applied Physics*, 92,10, 5930-5935, (2002).
(2) Zhang, S. and Zhang, X. Flipping a photonic shock wave. *Physics*, 2, 91, (2009).
3. How to decrease the significant loss in the mid-infrared band for a natural van der Waals (vdW) material? What is the physical mechanism? In addition, the typical value of the loss should be given for comparison.
4. The near-field case is discussed, how about the far-field case? Why do the authors concern about the near-field case?
5. The authors focus on the radiation distributions and angles. What is the intensity of the spoof reversed CR? Enhanced or weakened? How much? Why? Additionally, the efficiency was mentioned in the manuscript. However, how big is the efficiency? For the potential application (here on-chip mid-infrared nano-light sources), please predict the device efficiency. Compared with the conventional counterparts, what is the advantages?
6. Type I hyperbolic band was investigated. However, type II hyperbolic band, which has some advantages such as ultralow-loss and superior anisotropy shown in the Introduction section, was NOT employed for the spoof reversed CR. Why?
7. How about the coherence of the spoof reversed CR?

Reviewer #2:

Remarks to the Author:

In this manuscript, the authors demonstrate the reversed Cherenkov radiation (CR) at mid-infrared based on natural hyperbolic materials (e. g, MoO₃) through real-space images and reveal that the radiation angle and the quality factor of reversed CR can be increased by stacking the hBN layer on the MoO₃ surface. This work shows potential for long-sought-after applications in nano-optoelectronics, which can be accepted for publication after addressing a few important concerns with current manuscript.

1. MoO₃ is a natural van der Waals material, which can support both elliptical and hyperbolic phonon polaritons. Since it exhibits in-plane negative group-velocity dispersion for frequency range of 958-1010cm⁻¹, the authors demonstrate the reversed CR experimentally at 977cm⁻¹, whose IFC expresses as an ellipse. What about realizing reversed CR experimentally in other hyperbolic frequencies? As known, for MoO₃, hyperbolic frequencies also can exhibit reversed CR.
2. The authors mentions many times that the radiation efficiency of phonon polariton can be increased in the manuscript. However, there is no detailed data to support this claim. I am curious about the method to evaluate the radiation efficiency, from radiation intensity and radiation distance, or other standards? The authors should explain in details. In addition, in the end of the first paragraph on page 5, the authors claim that the radiation efficiency of the polaritonic reversed CR has been increased by extracting the interference fringes of phonon polaritons for reversed CR and no CR at the same position. However, in my opinion, this conclusion only can be drawn after the comparison between CR in MoO₃ and CR in other materials, instead of comparing with "no CR".

3. Please give the derivative process that how the IFC of hBN/MoO₃ can be obtained.
4. Spoof surface plasmon polaritons is an "artificial" surface plasmon polaritons, which are usually realized through some artificial microstructure, such as drilling a subwavelength square hole array in a metallic block. Therefore, in the manuscript, it could be better to change "spoof plasmon" as "surface plasmon".
5. Please add the scale for the IFCs of Fig.3 (c) and Fig.4 (b).
6. The authors said that "Since the plasmon has a positive dispersion while the phonon polaritons supported by MoO₃ have a negative dispersion, the wave vectors (k_e , k_{ph}) change in opposite directions when a dielectric material is inserted between the metal and MoO₃." If there are some references, please cite, if no, please prove.
7. There are some errors in this manuscript. For example, $\epsilon_{pz} \cos 2\beta + \epsilon_{py} \sin 2\beta$ should be changed as $\epsilon_{px} \cos 2\beta + \epsilon_{py} \sin 2\beta$. Please check it carefully.

Point-by-point response to the reviewers' comments on the manuscript entitled "Mid-infrared polaritonic reversed Cherenkov radiation in natural anisotropic crystals" (manuscript number: NCOMMS-22-48909A).

Dear Editor and Reviewers,

We thank the editor for handling our manuscript. We have revised the manuscript by carefully addressing all the comments and suggestions provided by the editor and reviewers, and have thoroughly checked our manuscript against the revision checklist. We believe that our revisions have significantly improved the quality and clarity of our manuscript.

We sincerely thank both Reviewers for their very positive feedback on our manuscript. Indeed, Reviewer 1 comments: " *It is an interesting topic for readers.*"; Reviewer 2 comments: " *This work shows potential for long-sought-after applications in nano-optoelectronics, which can be accepted for publication*".

We appreciate the insightful and constructive comments raised by the reviewers. Here, we have addressed all the comments. Our responses are shown in **blue**, and our revisions to the manuscript and supplementary material are indicated in **red**.

Reviewer #1

Comment: This paper reports a so-called reversed Cherenkov radiation (CR) in the mid-infrared band using natural anisotropic crystals. It is an interesting topic for readers. Here, the comments are listed below.

Reply: We appreciate the positive evaluation of the novelty of our work and the relevant suggestions to help clarify the technical issues for further improvement.

1. The authors claim that the reversed CR occurs in natural materials. This claim is NOT consistent with the Veselago's viewpoint.

Reply: We thank the reviewer for this comment.

We summarize Veselago's viewpoints first. And then, we will discuss how our claims align with Veselago's viewpoints:

Veselago's viewpoint I: In 1968, Veselago predicted that left-handed metamaterials with a negative refractive index (meaning both negative permittivity and permeability) would exhibit abnormal reversed Cherenkov radiation. This phenomenon allows for easy separation of the backwards-emitted wave from the emitting particles (V. G. Veselago, *Sov. Phys. Usp.* 10, 509 (1968)).

Veselago's viewpoint II: In his paper (V. G. Veselago, E. E. Narimanov, *Nat. Mater.* 5, 759–762 (2006)), Veselago commented that, for natural birefringent crystals with strong dielectric anisotropy, the electrical permittivity transverse to the surface is opposite to the in-plane component. In systems of reduced dimensionality (such as in a waveguide geometry), a dielectric anisotropy is sufficient to develop the full analogue of a negative index material — with both negative refraction for all angles and the phase velocity opposite to the energy flow (i.e., negative group-velocity dispersion).

According to Veselago's viewpoints I and II, the reversed CR typically occurs in media with a negative refractive index or negative group-velocity dispersion. We claim that the reversed CR can occur in natural materials, which have negative group-velocity dispersion (e.g., natural birefringent crystals like MoO_3). Therefore, our claims align well with Veselago's viewpoint II.

Changes made in the revised manuscript:

We thank the reviewer for this question, in response to which we have made the following changes in Lines 47-54 on Page 2 of the revised manuscript:

“Initially, it was theoretically predicted that reversed CR could be obtained in left-handed metamaterials (LHM) with a negative refractive index (i.e., simultaneously negative permittivity and permeability). Then reversed CR in the microwave band was experimentally demonstrated in artificial LHM, but the large loss of exquisite artificial

structures for optical frequency (i.e., $\sim 10^4 \text{ cm}^{-1}$ - 10^6 cm^{-1}) makes it difficult to realize reversed CR at this frequency range.

The recent emergence of two-dimensional van der Waals (vdW) materials (e.g., birefringent crystals), which have low-loss hyperbolic phonon polaritons with NGVD in the mid-infrared range, provides a competitive platform for reversed CR at the optical frequency”

We have cited the mentioned works as the new Refs. 11, 12 and 16:

“11 Veselago, V. G. Electrodynamics of substances with simultaneously negative electrical and magnetic permeabilities. *Sov. Phys. Usp.* 10, 504-509, (1968).

12 Pendry, J. B. & Smith, D. R. Reversing Light With Negative Refraction. *Phys. Today* 57, 37-43, (2004).

16 Veselago, V. & Narimanov, E. The left hand of brightness: past, present and future of negative index materials. *Nat. Mater.* 5, 759-762, (2006).”

2. CR is the electromagnetic radiation emitted by moving charges with a speed faster than the phase velocity of the electromagnetic wave inside a medium. However, in fact, the actual moving charged particles are not employed in the paper. The so-called reversed CR should be considered as analog of reversed CR or a special case of negative refraction not REAL reversed CR. This is because CR clearly shows the particle-wave interaction NOT wave (here light)-matter interaction. Please refer to the following two papers:

(1) Grbic, A. and Eleftheriades, G. V. Experimental verification of backward-wave radiation from a negative refractive index metamaterial. *Journal of Applied Physics*, 92, 10, 5930-5935, (2002).

(2) Zhang, S. and Zhang, X. Flipping a photonic shock wave. *Physics*, 2, 91, (2009).

Reply: We thank the reviewer for this suggestion and for providing these relevant references, which we have cited in the revised manuscript as Refs. 27, and 28. We agree with the Reviewer that the CR is directly excited by the moving charged particles, and

we experimentally measured the analogue of reversed CR, which has been modified in the revised manuscript.

We theoretically predicted the reversed CR in MoO₃ with the moving charged particles (Fig. 1c). However, this highly confined polaritonic reversed CR is hard to be experimentally characterized due to the lack of appropriate coupling and on-chip electron sources. To overcome this limitation, propagating plasmon mode or waveguide mode emulating the moving charged particles provide a way to study reversed CR, which is an analogue of reversed CR (*Nat. Nanotech.* 2015, 10, 804–809; *Nat. Commun.* 2017, 8, 1554; *Phys. Rev. Lett.* 2009, 103, 194801). In our work, we employed the plasmon mode in gold/silver nanowires to experimentally excite the analogue of the polaritonic reversed CR, which has been modified in the revised manuscript.

Changes made in the revised manuscript:

We thank the reviewer for this question, in response to which we have made the following change in Lines 111-115 and 118-123 on Pages 4-5 of the revised manuscript, as marked in red:

"The CR effect is electromagnetic radiation emitted when a charged particle passes through a dielectric medium at a speed greater than the phase velocity of light in the medium. Due to the challenges of combining a dielectric medium with fast-moving charged particles in a vacuum environment, an optical analog of CR has been developed as an ingenious way to study the physical effects of CR in depth. Based on the above theoretical analysis, we exhibit the imaging of in-plane reversed CR in MoO₃ in real space via scattering scanning near-field microscopy (s-SNOM), as illustrated in Fig. 2a. Compared with the far-field method of measuring CR, this near-field method can provide polaritonic reversed CR distribution with approximately 10 nm spatial resolution, such as the radiation wavefront and radiation angle of CR. In order to excite the polaritonic reversed CR, plasmon (i.e., a dynamic charge density wave), which is analogous to a superluminal charge, propagating in a one-dimensional metal nanowire, serves as the radiation emitter."

We have cited the noted previous works as the new Refs. 26-28, as marked in red:

“26 Lyu, Z. et al. Compact reversed Cherenkov radiation oscillator with high efficiency. *Appl. Phys. Lett.* 120, 053501 (2022).

27 Grbic, A. & Eleftheriades, G. V. Experimental verification of backward-wave radiation from a negative refractive index metamaterial. *J. Appl. Phys.* 92, 5930-5935, (2002).

28 Zhang, S. & Zhang, X. Flipping a photonic shock wave. *Physics* 2, 91, (2009).”

3. How to decrease the significant loss in the mid-infrared band for a natural van der Waals (vdW) material? What is the physical mechanism? In addition, the typical value of the loss should be given for comparison.

Reply: We thank the reviewer for raising this important question. In natural van der Waals (vdW) materials, losses in the mid-infrared band are mainly caused by dielectric and scattering losses due to phonons and impurities. To decrease these losses, we can explore the following pathways (as shown in Table R1 of this reply letter): (1) using the isotope-enriched crystals to reduce the isotope-induced scattering of phonons (*Nano Lett.* 2022, 22, 10208; *Nat. Mater.* 2018, 17, 134); (2) cooling the system to cryogenic temperatures to minimize temperature-dependent phonon scattering (*Nano Lett.*, 2021, 21, 5767; *Appl. Phys. Lett.*, 2022, 120, 161101.); and (3) changing the surrounding dielectric environment to reduce the dielectric losses and interface scattering (*Nano Lett.* 2019, 19, 1009; *Nat. Commun.*, 2022, 13, 1465; *Appl. Phys. Lett.* 2022, 120, 113101; *Adv. Opt. Mater.*, 2022, 10, 2102057).

Here is the summarization of the typical loss (lifetime or FOM) for MoO₃:

Methods Frequency	Natural MoO ₃ /SiO ₂ ¹	Isotope- enriched MoO ₃ /Si ²	MoO ₃ /SiO ₂ at cryogenic temperature ^{3,4}	Changing dielectric environment	
				Suspended MoO ₃ ^{5,6}	MoO ₃ /Gold ⁷
Type II hyperbolic band	~1.9 ps (FOM~2.4)	~2.3 ps	~2.2 ps	FOM~2.8- 3.2	FOM~4.5

Type I hyperbolic band	~8 ps (FOM~3)	~13.9 ps	FOM~4.1	FOM~3.5- 4.2	FOM~3.2
------------------	----------	---------	-----------------	---------

Table R1: Typical losses of polaritons in MoO₃. There are slight differences in the polaritonic properties under different sample thicknesses and incident light frequencies. FOM: Figure of Merit = $Re\{q\}/2\pi Im\{q\}$, $Re\{q\}$ is the real part of momentum (q), $Im\{q\}$ is the imaginary part of q .

4. The near-field case is discussed, how about the far-field case? Why do the authors concern about the near-field case?

Reply: We thank the reviewer for asking this intriguing question. As far as we know, it is challenging to collect reversed CR signals in the far field because of the low coupling efficiency of the in-plane CR signal to the out-of-plane space. In the case of polaritonic reversed CR, the in-plane propagation distance is $\sim 10^3$ nm, while the out-of-plane evanescent length is $\sim 10^2$ nm. As a result, it is difficult to find a suitable coupler in far-field technology to detect the polaritonic reversed CR signal.

In recent years, we only focus on the near-field study of the polaritonic reversed CR in MoO₃, which has the potential for nanoscale on-chip light sources. In our previous work (*Nat. Mater.*, 2021, 20, 43), we investigated phonon polaritons in h-BN using electron energy loss spectroscopy (EELS) and observed the Cherenkov physical effects of phonon polaritons through aloof excitation of a high-energy electron beam (point M in Fig. R1). However, the real-space field distribution (e.g., radiation wavefronts, and radiation angle) of this CR cannot be further analyzed due to the low energy resolution of the EELS technology. Compared with EELS, scattering scanning near-field microscopy (s-SNOM) can not only satisfy the momentum compensation requirements for reversed CR excitation but also spatially map the propagating wavefront of polaritonic reversed CR in the near field (with ~ 10 nm resolution). Then, by capturing real-space images of MoO₃ under different conditions (e.g., varied thicknesses, different polarization directions of light, broken in-plane symmetry, etc),

the radiation distributions and angles of reversed CR can be well investigated in the near-field.

Figure R1. EELS spectra were acquired at two positions: inside the sample (magenta dotted line) and outside the sample (cyan dotted line). For EELS spectra obtained via aloof excitation, the Cherenkov physical effect is clearly present, but it is difficult to observe the accurate imaging distribution of CR in real-space (details in our previous work *Nat. Mater.*, 2021, 20, 43).

Changes made in the revised manuscript:

We thank the reviewer for this question, in response to which we have made the following change in lines 118-123 on Pages 4-5 of the revised supplementary information, as marked in red:

“Compared with the far-field method of measuring CR, this near-field method can provide polaritonic reversed CR distribution with approximately 10 nm spatial resolution, such as the radiation wavefront and radiation angle of CR. In order to excite the polaritonic reversed CR, plasmon (i.e., a dynamic charge density wave), which is analogous to a superluminal charge, propagating in a one-dimensional metal nanowire, serves as the radiation emitter.”

5. The authors focus on the radiation distributions and angles. (5.1) What is the intensity of the spoof reversed CR? Enhanced or weakened? How much? Why? (5.2) Additionally, the efficiency was mentioned in the manuscript. However, how big is the efficiency? For the potential application (here on-chip mid-infrared nano-light sources),

please predict the device efficiency. (5.3) Compared the conventional counterparts, what is the advantages?

Reply: We sincerely thank the reviewer for his/her positive comments.

(5.1): The intensity of polaritonic reversed CR is proportional to the $|E_z|^2$ (e.g., $|E_z|$ is the electric field in Fig.2d). By extracting the electromagnetic field distribution, we find that the intensity of polaritonic reversed CR propagating on the MoO₃ is enhanced, which is ~ 10 times (e.g., $|E_z/E_{z0}|^2$) higher than the intensity of the supporting substrate ($|E_{z0}|^2$). This is mainly corresponding to the great wave vector compression ability ($k_{ph}/k_0 \sim 20.8$ in our work) of phonon polaritons.

Figure R2. The simulated electromagnetic energy distribution of polaritonic reversed CR on MoO₃. The frequency is 980 cm⁻¹. The thickness of MoO₃ is about 400 nm. Scale bar: 1 μ m.

(5.2): In our work, we found that the radiation efficiency of the CR is approximately 3.8% in MoO₃ and approximately 5.1% in hBN/MoO₃ (see details in Fig.R3), respectively. The radiation efficiency of CR is the radiation rate at which charged particles (i.e., metal nanowire plasmon in our work) radiate to the polaritonic reversed CR wavefront (*Nano Lett.*, 2020, 20, 2770). The radiation rate (η) of the polaritonic CR is determined by $k_e/(k_{ph} - k_e)$, where k_e and k_{ph} are the wave vectors of plasmons in the metal nanowires and excited phonon polaritons, respectively. We extract the k_{ph} of the reversed polaritonic CR in both MoO₃ and hBN/MoO₃ by measuring their wavelengths at the frequency of 977 cm⁻¹ (details in Figure R3). The k_e is adapted as the wave vector of light in free space since there is nearly no

wavelength compression for Au/Ag plasmons in the mid-infrared band. Thus, the predicted η is $\sim 3.8\%$ in MoO_3 and $\sim 5.1\%$ in hBN/MoO_3 , respectively.

Figure R3. **a**, The schematic illustration of reversed CR in the air/ MoO_3 and hBN/MoO_3 heterostructures. The red dashed lines represent the coupling between plasmons and phonon polaritons, which determine the corresponding radiation efficiency. Experimental polaritonic reversed Cherenkov radiation of air/ MoO_3 (**b**) hBN/MoO_3 (**c**) with excitation frequency of 977 cm^{-1} . The thickness of MoO_3 is about 280 nm, and the thickness of hBN is about 7 nm. Scale bars: 2 μm . The extracted fringes of phonon polaritons in (**d**), MoO_3 and (**e**), hBN/MoO_3 with an excitation frequency of 977 cm^{-1} . Experiment data: points. Fitting data: lines.

(5.3): Compared to conventional counterparts (*Phys. Rev. Lett.*, 2009, 103, 194801; *Nat Commun* 2017, 8, 14901; *Phys. Rev. Lett.*, 2019, 122, 014801; *Appl. Phys. Lett.* 2022, 120, 053501), our polaritonic reversed CR has the following advantages:

(I) Optical frequency polaritonic reversed CR: The frequency of reversed CR in conventional metamaterials is usually in the microwave frequency. This is because metamaterials are made from a repeated arrangement of multiple elements like metals and plastics, which are much smaller than the working wavelengths. It is still a challenge for existing three-dimensional processing technologies to achieve such metamaterials for the optical frequency reversed CR. In contrast, the natural crystal

MoO₃ can directly excite polaritonic reversed CR with tunable radiation rates and angles via layer thickness, the direction of the crystal axis, etc.

(II) Nanoscale in-plane polaritonic reversed CR: The conventional reversed CR is usually three-dimensional and emitted into out-of-plane space. In contrast, polaritonic reversed CR maintains the great wave vector compression ability of phonon polaritons, which can be highly confined on the MoO₃ surface to the formation of on-chip 2D CR.

Changes made in the revised manuscript:

We thank the reviewer for this question, in response to which we have made the following change in lines 189-196 on Page 7 and lines 220-223 on Page 8 of the revised manuscript, as marked in red:

“To further increase the radiation angle (θ_k) and improve the radiation efficiency ($\eta \approx |k_e|/(|k_{ph}| - |k_e|)$) of the polaritonic reversed CR, one can increase the plasmonic wave vector (k_e) or reduce the phonon polaritonic wave vector (k_{ph}). A higher CR efficiency can be achieved by increasing the coupling between plasmons and phonon polaritons (i.e., decreasing the momentum mismatch between the large k_{ph} and low k_e), which makes it easier for plasmons (source) to transfer energy and momentum to phonon polaritons (the excited reversed CR). The increase of k_{ph} and decrease of k_e can be realized by increasing the environmental dielectric since the plasmon has a positive dispersion while the phonon polaritons supported by MoO₃ have a negative dispersion”

“Furthermore, benefitting from the low loss of Cherenkov phonon polaritons, the experimentally observed reversed CR exhibits good Q factors and radiation efficiency η , such as $Q \approx 10$, $\eta \approx 3.8\%$ in air/MoO₃ and $Q \approx 16$, $\eta \approx 5.1\%$ in hBN/MoO₃ (details in Fig. S5c of Supplementary information).”

We have also added Fig. S5c on Page 7 (lines 136-139) of the revised supplementary information, as marked in red:

“

Supplementary Figure S5. Quality factors of hBN/MoO₃ heterostructure and MoO₃. **c**, The extracted interference fringes of phonon polaritons in MoO₃ and hBN/MoO₃ (Fig. 4c) with an excitation frequency of 977 cm⁻¹. Experiment data: points. Fitting data: lines. The thickness of MoO₃ is 280 nm.”

6. Type I hyperbolic band was investigated. However, type II hyperbolic band, which has some advantages such as ultralow-loss and superior anisotropy shown in the Introduction section, was NOT employed for the spoof reversed CR. Why?

Reply: We thank the reviewer for raising this important question. In our investigation of polaritonic reversed CR, we focused on the type I hyperbolic band, which has a higher frequency than the type II hyperbolic band (Fig. R4a), which is easier to be investigated experimentally. The polaritonic reversed CR in the type II hyperbolic band requires a larger electron wave vector (e.g., $k_{e2} > 47k_0$ in Fig.R4c) to be excited. Therefore, current research in frequency band II is limited to theoretical simulations.

Specifically, for the type I hyperbolic band, the electron wavevector ($k_{e1} < 23k_0$) located within the elliptical IFC can excite in-plane polaritonic CR, as shown in Fig. R4b. For the type II hyperbolic band, the electron wavevector ($k_{e2} > 47k_0$) located outside the hyperbolic IFC can excite polaritonic CR, as shown in Fig. R4c. In our s-SNOM experiment, plasmons in gold/silver nanowires were utilized to imitate fast-moving charged particles which can provide a relatively small wavevector ($k_e < 5k_0$) due to their weak compression ability in the mid-infrared band (*Sci. Adv.*, 2022, 8, 29). Thus, only the reversed CR in the type I hyperbolic band was studied.

Figure R4. **a**, The dispersion of MoO₃ phonon polariton with type I and type II hyperbolic bands. (Reproduced Figure in *Nano Lett.*, 2022, 22, 10208) **b-c**, Calculated IFCs of MoO₃ phonon polariton with type I and type II hyperbolic bands. The thickness of MoO₃ is about 300 nm, and the frequency is **(b)** 980 cm⁻¹ and **(c)** 920 cm⁻¹. For the type I hyperbolic band, the electron wavevector (k_{e1}) located within the elliptical IFC can excite polaritonic CR. For the type II hyperbolic band, the electron wavevector (k_{e2}) located out the hyperbolic IFC can excite polaritonic CR.

Finally, we want to claim that both type I and type II hyperbolic bands offer advantages such as ultralow-loss and superior anisotropy. In our previous statement, we may mislead the reader that the type II hyperbolic band is better than the type I hyperbolic band. We have carefully rewritten the statement in the revised manuscript (Lines 57-61 on page 2), which is stated as follows: “**In particular, it has been shown that phonon polaritons in α -MoO₃ (MoO₃), which have optical frequency response, ultralow-loss, and superior anisotropy, provide an ideal platform for realizing reversed CR at the optical frequency and for achieving multidimensional manipulation of the reversed CR (e.g., in the Reststrahlen band of 820-972 cm⁻¹).**”

Changes made in the revised manuscript:

We appreciate the reviewer for bringing up this concern. As a response, we have revised the statement in lines 57-61 on Page 2 of the revised manuscript, as marked in red:

“In particular, it has been shown that phonon polaritons in α -MoO₃ (MoO₃), which have optical frequency response, ultralow-loss, and superior anisotropy, provide an ideal platform for realizing reversed CR at the optical frequency and for achieving multidimensional manipulation of the reversed CR (e.g., in the Reststrahlen band of 820-972 cm⁻¹).”

7. How about the coherence of the spoof reversed CR?

Reply: We thank the reviewer for asking this question. Polaritonic reversed CR maintains the coherent performance of phonon polaritons (*Nano Lett.*, 2020, 20, 2770; *Physica B* 2002, 316, 55; *Science*, 2021, 372, 1181). Furthermore, our earlier research (*Adv. Func. Mater.*, 2019, 29, 1904662.) proves that the real space wavefront formation process of phonon polariton is an interference process, which directly reflects its good coherence performance in s-SNOM experiment. Finally, in order to quantify its spatial coherence, we have fitted the coherence length (L) of the polaritonic reversed CR at different frequencies (Fig. R5), which can reach $\sim 1.7 \mu\text{m}$.

Figure R5 **a,c** Experimental reversed Cherenkov radiation of MoO₃ phonon polaritons with an excitation frequency of 973 cm⁻¹ and 971 cm⁻¹. Scale bars: 1 μm . **b,d** The extracted fringes of phonon polaritons in (**a,c**) along black dashed arrows. Experiment data: points. Fitting data: red lines.

Changes made in the revised manuscript:

We thank the reviewer for asking this question, in response to which we have made the following change in lines 140-143 on Page 5 of the main text, as marked in red:

“This is because the polaritonic reversed CR is excited by the propagating plasmons in the Au/Ag nanowire whose strength is enhanced compared to the incident light. In addition, polaritonic reversed CR maintains the coherent performance of phonon polaritons.”

We have cited the noted previous works as the new Refs. 31 and 32:

“31 Wahlstrand, J., Stevens, T., Kuhl, J. & Merlin, R. Coherent phonon–polaritons and subluminal Cherenkov radiation. *Physica B: Condensed Matter* 316, 55-61, (2002).

32 Kurman, Y. et al. Spatiotemporal imaging of 2D polariton wave packet dynamics using free electrons. *Science* 372, 1181-1186, (2021).”

Reviewer #2

Comment: In this manuscript, the authors demonstrate the reversed Cherenkov radiation (CR) at mid-infrared based on natural hyperbolic materials (e. g, MoO₃) through real-space images and reveal that the radiation angle and the quality factor of reversed CR can be increased by stacking the hBN layer on the MoO₃ surface. This work shows potential for long-sought-after applications in nano-optoelectronics, which can be accepted for publication after addressing a few important concerns with current manuscript.

Reply: We sincerely thank the reviewer for his/her positive comments and the recommendation for publication.

1. MoO₃ is a natural van der Waals material, which can support both elliptical and hyperbolic phonon polaritons. Since it exhibits in-plane negative group-velocity dispersion for frequency range of 958-1010 cm⁻¹, the authors demonstrate the reversed CR experimentally at 977 cm⁻¹, whose IFC expresses as an ellipse. What about realizing reversed CR experimentally in other hyperbolic frequencies? As known, for MoO₃, hyperbolic frequencies also can exhibit reversed CR.

Reply: We thank the reviewer for bringing up this important question. In addition to the type I hyperbolic band studied in our work, the polaritonic reversed CR has also been theoretically predicted in the type II hyperbolic band (*Opt. Lett.*, 2022, 47, 2458).

Specifically, when the in-plane extreme anisotropy satisfies $\partial\omega/\partial k < 0$, the in-plane polaritonic reversed CR in type II can be realized. Fig. R6a,b shows two typical IFC in the type I and type II hyperbolic bands, respectively, to schematically illustrate the conditions required to excite the in-plane polaritonic reversed CR. Specifically, for the type I hyperbolic band, the electron wavevector ($k_{e1} < 23k_0$) located within the elliptical IFC can excite in-plane polaritonic CR, as shown in Fig. R6a. For the type II hyperbolic band, the electron wavevector ($k_{e2} > 47k_0$) located outside the hyperbolic IFC can excite polaritonic CR, as shown in Fig. R6b. Thus, realizing in-plane reversed CR in the type II band requires a much larger electron wavevector than that in the type I band, which is challenging to achieve experimentally

In our s-SNOM measurement of the in-plane polaritonic reversed CR, we employed gold/silver nanowires to imitate a fast-moving charged particle. However, due to its high radiation loss in the infrared band, most isolated metal nanowires have a small compression ability ($k_e < 5 \times k_0$) and can only excite in-plane reversed CR in the type I hyperbolic band in experiments (*Sci. Adv.*, 2022, 8, 29). In the future, we believe it may be possible to experimentally realize type II reversed CR by providing the required wave vector compensation.

Figure R6. Calculated IFCs of MoO₃ phonon polariton with type I and type II hyperbolic bands, respectively. The thickness of MoO₃ is about 300 nm, and the frequency is (a) 980 cm⁻¹ and (b) 920 cm⁻¹. For the type I hyperbolic band, the electron wavevector (k_{e1}) located within the elliptical IFC can excite polaritonic CR; For the type II hyperbolic band, the electron wavevector (k_{e2}) located outside the hyperbolic

IFC can excite polaritonic CR.

2. The authors mentions many times that the radiation efficiency of phonon polariton can be increased in the manuscript. (2.1) However, there is no detailed data to support this claim. I am curious about the method to evaluate the radiation efficiency, from radiation intensity and radiation distance, or other standards? The authors should explain in details. (2.2) In addition, in the end of the first paragraph on page 5, the authors claim that the radiation efficiency of the polaritonic reversed CR has been increased by extracting the interference fringes of phonon polaritons for reversed CR and no CR at the same position. However, in my opinion, this conclusion only can be drawn after the comparison between CR in MoO₃ and CR in other materials, instead of comparing with “no CR”.

Reply: We thank the reviewer for this comment.

(2.1) In our study, the radiation efficiency of the polaritonic reversed CR was determined to be approximately 3.8% in MoO₃ and 5.1% in hBN/MoO₃ (see details in Fig.R7). The radiation efficiency of the CR refers to the rate at which charged particles (such as metal nanowire plasmons in our work) radiate to the polaritonic reversed CR wavefront (*Nano Lett.*, 2020, 20, 2770). The coupling between plasmons and phonon polaritons is dependent on momentum matching, and the radiation rate (η) of the polaritonic CR is predicted by $k_e/(k_{ph} - k_e)$, where k_e and k_{ph} are the wave vectors of plasmons in the metal nanowires and excited phonon polaritons in MoO₃, respectively. We extract the k_{ph} of the polaritonic reversed CR in both MoO₃ and hBN/MoO₃ by measuring their wavelengths at the frequency of 977 cm⁻¹ (see details in Figure R9). The k_e is adapted as the wave vector of light in free space since there is nearly no wavelength compression for Au/Ag plasmons in the mid-infrared band. Thus, the η was calculated to be ~3.8% in MoO₃ and ~5.1% in hBN/MoO₃, respectively.

(2.2) We agree with the reviewer’s suggestion that polaritonic reversed CR and no CR should not be compared together as there is no concept of radiation efficiency without CR. Therefore, we have carefully rewritten the statement in the revised manuscript (Lines 140-143 on page 5) from: “This is because the excited plasmons

propagating longitudinally along the nanowire can couple to the phonon polariton, which increases radiation efficiency of the polaritonic reversed CR.” to “This is because the polaritonic reversed CR is excited by the propagating plasmons in the Au/Ag nanowire whose strength is enhanced compared to the incident light. In addition, polaritonic reversed CR maintains the coherent performance of phonon polaritons.”

Furthermore, we also compare the two cases of polaritonic reversed CR in MoO₃ and in hBN/MoO₃ heterostructure (Fig.4). As shown in Fig.R7, the radiation efficiency in the hBN/MoO₃ heterostructure (~5.1%) is higher than that in MoO₃ (~3.8%). This is due to the fact that the constructed heterostructure system effectively reduces the wave vector of MoO₃ phonon polaritons. This reduction in wavevector can result in a decrease in the momentum mismatch between plasmons and phonon polaritons, leading to enhanced coupling and a higher radiation efficiency of the polaritonic reversed CR.

Figure R7. **a**, Experimental data of reversed Cherenkov radiation of MoO₃ phonon polaritons with an excitation frequency of 977 cm⁻¹ when $\beta \sim 90^\circ$. The thickness of MoO₃ is about 280 nm. Scale bars: 2 μm . **b**, Experimental reversed Cherenkov radiation of hBN/MoO₃ phonon polaritons with an excitation frequency of 977 cm⁻¹ when $\beta \sim 90^\circ$. The thickness of MoO₃ is about 280 nm, and the thickness of hBN is about 7 nm. Scale bars: 1.5 μm . The extracted fringes of phonon polaritons in **c**, MoO₃ and **d**, hBN/MoO₃

with an excitation frequency of 977 cm^{-1} . Experiment data: points. Fitting data: lines.

Changes made in the revised manuscript:

We thank the reviewer for raising this question, in response to which we have made the following changes in lines 189-196 on Page 7 and lines 220-223 on Page 8 of the revised manuscript, as marked in red:

“To further increase the radiation angle (θ_k) and improve the radiation efficiency ($\eta \approx |k_e|/(|k_{ph}| - |k_e|)$) of the polaritonic reversed CR, one can increase the plasmonic wave vector (k_e) or reduce the phonon polaritonic wave vector (k_{ph}). A higher CR efficiency can be achieved by increasing the coupling between plasmons and phonon polaritons (i.e., decreasing the momentum mismatch between the large k_{ph} and low k_e), which makes it easier for plasmons (source) to transfer energy and momentum to phonon polaritons (the excited reversed CR). The increase of k_{ph} and decrease of k_e can be realized by increasing the environmental dielectric constant. This is because the plasmon has a positive dispersion while the phonon polaritons supported by MoO_3 have a negative dispersion.”

“Furthermore, benefitting from the low loss of Cherenkov phonon polaritons, the experimentally observed reversed CR exhibits good Q factors and radiation efficiency η , such as $Q \approx 10$, $\eta \approx 3.8\%$ in air/ MoO_3 and $Q \approx 16$, $\eta \approx 5.1\%$ in hBN/ MoO_3 (details in Fig. S5c of Supplementary information).”

We have made the following change in lines 140-143 on page 5 of the revised manuscript:

“This is because the polaritonic reversed CR is excited by the propagating plasmons in the Au/Ag nanowire whose strength is enhanced compared to the incident light. In addition, polaritonic reversed CR maintains the coherent performance of phonon polaritons”.

We have also added Fig. S5c on Page 7 (lines 136-139) of the revised supplementary information, as marked in red:

“

Supplementary Figure S5. Quality factors of hBN/MoO₃ heterostructure and MoO₃. **c**, The extracted interference fringes of phonon polaritons in MoO₃ and hBN/MoO₃ (Fig. 4c) with an excitation frequency of 977 cm⁻¹. Experiment data: points. Fitting data: lines. The thickness of MoO₃ is 280 nm.”

3. Please give the derivative process that how the IFC of hBN/MoO₃ can be obtained.
 Reply: We thank the reviewer’s suggestion. We have carried out the detailed derivative process of IFC below, which has been added in the revised supplementary information.

The hBN/MoO₃ can be treated as a laterally infinitely layered medium consisting of four layers (Fig. R8): $z > 0$ (air, layer $j = 1$), $-d_1 < z < 0$ (hBN, layer $j = 2$), $-(d_1 + d_2) < z < -d_1$ (MoO₃, layer $j = 3$), and $z < -(d_1 + d_2) = -d$ (SiO₂, layer $j = 4$). Without loss of generality, we assume polaritons in hBN/MoO₃ propagate along the $k_{in-plane}$ direction in the xy plane. In the xyz coordinates, we have

$$k_x = k_{in-plane} \cdot \sin(\phi), \quad k_y = k_{in-plane} \cdot \cos(\phi), \quad (S1)$$

where ϕ represents the angle between $\vec{k}_{in-plane}$ and the x axis. Thus the dielectric function of MoO₃ along the direction of $\vec{k}_{in-plane}$ can be expressed as $\epsilon_{mt} = \epsilon_{mx} \cos^2(\phi) + \epsilon_{my} \sin^2(\phi)$.

Polaritons in MoO₃ are generally hybrid transverse magnetic-transverse electrical (TM-TE) eigenmodes, but the intensity of the TE mode is much weaker than the TM mode (TM: E_x, H_y, E_z) (see *Nat. Mater.* 2020, 19, 1307; *Nanoscale*, 2021, 13, 4845). The transverse magnetic field in the hBN/MoO₃ heterostructure can be expressed as $\vec{H}_y = \vec{e}_y H_y(z) e^{(iqx - i\omega t)}$, which can satisfy the wave equation $\nabla^2 \vec{H} + \vec{\epsilon} k_0^2 \vec{H} = \nabla(\nabla \cdot \vec{H})$. Thus

$$\frac{\partial^2 \vec{H}_y}{\partial z^2} + (\epsilon_{a,s}^{(j)} k_0^2 - q^2) \vec{H}_y = 0, \quad (\text{S2})$$

$$\frac{\partial^2 \vec{H}_y}{\partial z^2} + \left(\epsilon_t^{(j)} k_0^2 - \frac{\epsilon_t^{(j)}}{\epsilon_z^{(j)}} q^2 \right) \vec{H}_y = 0, \quad (\text{S3})$$

There are evanescent fields in both the air layer and the substrate layer, so the interface can satisfy $\epsilon_{a,s}^{(j)} k_0^2 - q^2 < 0$, $j = 0, 3$ and $\epsilon_t^{(j)} k_0^2 - (\epsilon_t^{(j)} / \epsilon_z^{(j)}) q^2 > 0$, $j = 1, 2$. In the four-layer structure, the distribution of H_y satisfies:

$$H_y(z) = \begin{cases} (P_1 + P_2) e^{-Q_a z}, & z > 0 \\ P_1 e^{ik_z^{(1)} z} + P_2 e^{-ik_z^{(1)} z}, & -d_1 < z \leq 0 \\ P_3 e^{ik_z^{(2)} z} + P_4 e^{-ik_z^{(2)} z}, & -d < z \leq -d_1 \\ (P_3 e^{-ik_z^{(2)} d} + P_4 e^{ik_z^{(2)} d}) e^{-Q_s(z+d)}, & z \leq -d \end{cases} \quad (\text{S4})$$

According to $\nabla \times H = i\omega \epsilon_0 \vec{\epsilon} E$, Thus

$$E_x(z) = \begin{cases} \frac{iQ_a}{\omega \epsilon_0 \epsilon_a} (P_1 + P_2) e^{-Q_a z}, & z > 0 \\ \frac{1}{\omega \epsilon_0 \epsilon_t^{(1)}} (P_1 e^{ik_z^{(1)} z} + P_2 e^{-ik_z^{(1)} z}), & -d_1 < z \leq 0 \\ \frac{1}{\omega \epsilon_0 \epsilon_t^{(2)}} (P_3 e^{ik_z^{(2)} z} + P_4 e^{-ik_z^{(2)} z}), & -d < z \leq -d_1 \\ \frac{-iQ_s}{\omega \epsilon_0 \epsilon_s} (P_3 e^{-ik_z^{(2)} d} + P_4 e^{ik_z^{(2)} d}) e^{-Q_s(z+d)}, & z \leq -d \end{cases} \quad (\text{S5})$$

Where $Q_{a,s} = \sqrt{-\epsilon_{a,s} k_0^2 + q^2}$, and $k_z^{(1,2)} = \sqrt{\epsilon_t^{(1,2)} k_0^2 - (\epsilon_t^{(1,2)} / \epsilon_z^{(1,2)}) q^2}$.

Considering that the tangential components of \vec{E} and \vec{H} are continuous at the interface,

$$\begin{cases} E_x^{(0)} = E_x^{(1)}, & H_y^{(0)} = H_y^{(1)}, & z = 0 \\ E_x^{(1)} = E_x^{(2)}, & H_y^{(2)} = H_y^{(3)}, & z = -d_1 \\ E_x^{(2)} = E_x^{(3)}, & H_y^{(2)} = H_y^{(3)}, & z = -d \end{cases} \quad (\text{S6})$$

Thus the coefficients (P_1-P_4) can satisfy the $M \cdot (P_1, P_2, P_3, P_4)^T = 0$,

$$M = \begin{pmatrix} \frac{iQ_a}{\varepsilon_a} - \frac{k_z^{(1)}}{\varepsilon_t^{(1)}} & \frac{iQ_a}{\varepsilon_a} + \frac{k_z^{(1)}}{\varepsilon_t^{(1)}} & 0 & 0 \\ \frac{k_z^{(1)}}{\varepsilon_t^{(1)}} e^{-ik_z^{(1)}d_1} & -\frac{k_z^{(1)}}{\varepsilon_t^{(1)}} e^{ik_z^{(1)}d_1} & -\frac{k_z^{(2)}}{\varepsilon_t^{(2)}} e^{-ik_z^{(2)}d_1} & \frac{k_z^{(2)}}{\varepsilon_t^{(2)}} e^{ik_z^{(2)}d_1} \\ e^{-ik_z^{(1)}d_1} & e^{ik_z^{(1)}d_1} & -e^{-ik_z^{(2)}d_1} & e^{ik_z^{(2)}d_1} \\ 0 & 0 & \left(\frac{iQ_s}{\varepsilon_s} + \frac{k_z^{(2)}}{\varepsilon_t^{(2)}}\right) e^{-ik_z^{(2)}d} & \left(\frac{iQ_s}{\varepsilon_s} - \frac{k_z^{(2)}}{\varepsilon_t^{(2)}}\right) e^{ik_z^{(2)}d} \end{pmatrix}$$

When the determinant $\det\{M\} = 0$, it is guaranteed that there are non-zero solutions for the amplitude coefficients (P_1-P_4) . Specifically, the dispersion relation can be stated as,

$$e^{2n_1} = -\frac{[(n_2-n_3)(n_3-n_4)(n_4+n_5)]e^{ik_z^{(1)}d_1} + [(n_2+n_3)(n_3+n_4)(n_4+n_5)]e^{-ik_z^{(1)}d_1}}{[(n_2-n_3)(n_3+n_4)(n_4-n_5)]e^{ik_z^{(1)}d_1} + [(n_2+n_3)(n_3-n_4)(n_4-n_5)]e^{-ik_z^{(1)}d_1}} \quad (S7)$$

Where $n_1 = -ik_z^{(2)}d_1 + ik_z^{(2)}d$, $n_2 = \frac{iQ_a}{\varepsilon_a}$, $n_3 = \frac{k_z^{(1)}}{\varepsilon_t^{(1)}}$, $n_4 = \frac{k_z^{(2)}}{\varepsilon_t^{(2)}}$, and $n_5 = \frac{iQ_s}{\varepsilon_s}$.

Therefore, the IFC of polaritons in hBN/MoO₃ can be obtained by solving equation (S7) with equation (S1). In addition, previous works (see *Nature* 2020, 582, 209; *Nat. Mater.* 2020, 19, 1307; *Nano Lett.*, 2020, 20, 5301; *Nano Lett.*, 2020, 20, 5323; *Nanoscale*, 2021, 13, 4845), which are cited in the revised manuscript, demonstrated the derivative process of the IFC of phonon polaritons in MoO₃-based bilayer heterostructure.

Figure R8. Schematic illustration of hBN/MoO₃ heterostructure which is treated as a laterally infinitely layered medium consisting of four layers.

Changes made in the revised manuscript:

We have followed the reviewer's suggestion and added Fig. R8 to the revised Supplementary Information as the new Fig. S11. We have also added Supplementary

Note 3: The IFC of hBN/MoO₃ phonon polaritons on Pages 3-5 of the revised Supplementary information, as marked in red:

“The hBN/MoO₃ can be treated as a laterally infinitely layered medium consisting of four layers (Supplementary Fig. S11): $z > 0$ (air, layer $j = 1$), $-d_1 < z < 0$ (hBN, layer $j = 2$), $-(d_1 + d_2) < z < -d_1$ (MoO₃, layer $j = 3$), and $z < -(d_1 + d_2) = -d$ (SiO₂, layer $j = 4$). Without loss of generality, we assume polaritons in hBN/MoO₃ propagate along the $k_{in-plane}$ direction in the xy plane. In the xyz coordinates, we have

$$k_x = k_{in-plane} \cdot \sin(\phi), \quad k_y = k_{in-plane} \cdot \cos(\phi), \quad (S4)$$

where ϕ represents the angle between $\vec{k}_{in-plane}$ and the x axis. Thus the dielectric function of MoO₃ along the direction of $\vec{k}_{in-plane}$ can be expressed as $\varepsilon_{mt} = \varepsilon_{mx} \cos^2(\phi) + \varepsilon_{my} \sin^2(\phi)$.

Polaritons in MoO₃ are generally hybrid transverse magnetic-transverse electrical (TM-TE) eigenmodes, but the intensity of the TE mode is much weaker than the TM mode (TM: E_x, H_y, E_z). The transverse magnetic field in the hBN/MoO₃ heterostructure can be expressed as $\vec{H}_y = \vec{e}_y H_y(z) e^{(iqx - i\omega t)}$, which can satisfy the wave equation $\nabla^2 \vec{H} + \vec{\varepsilon} k_0^2 \vec{H} = \nabla(\nabla \cdot \vec{H})$. Thus

$$\frac{\partial^2 \vec{H}_y}{\partial z^2} + \left(\varepsilon_{a,s}^{(j)} k_0^2 - q^2 \right) \vec{H}_y = 0, \quad (S5)$$

$$\frac{\partial^2 \vec{H}_y}{\partial z^2} + \left(\varepsilon_t^{(j)} k_0^2 - \frac{\varepsilon_t^{(j)}}{\varepsilon_z^{(j)}} q^2 \right) \vec{H}_y = 0, \quad (S6)$$

There are evanescent fields in both the air layer and the substrate layer, so the interface can satisfy $\varepsilon_{a,s}^{(j)} k_0^2 - q^2 < 0$, $j = 0, 3$ and $\varepsilon_t^{(j)} k_0^2 - (\varepsilon_t^{(j)} / \varepsilon_z^{(j)}) q^2 > 0$, $j = 1, 2$. In the four-layer structure, the distribution of H_y satisfies:

$$H_y(z) = \begin{cases} (P_1 + P_2) e^{-Q_a z}, & z > 0 \\ P_1 e^{ik_z^{(1)} z} + P_2 e^{-ik_z^{(1)} z}, & -d_1 < z \leq 0 \\ P_3 e^{ik_z^{(2)} z} + P_4 e^{-ik_z^{(2)} z}, & -d < z \leq -d_1 \\ (P_3 e^{-ik_z^{(2)} d} + P_4 e^{ik_z^{(2)} d}) e^{-Q_s(z+d)}, & z \leq -d \end{cases} \quad (S7)$$

According to $\nabla \times H = i\omega \varepsilon_0 \vec{\varepsilon} E$, Thus

$$E_x(z) = \begin{cases} \frac{iQ_a}{\omega\varepsilon_0\varepsilon_a} (P_1 + P_2)e^{-Q_az}, & z > 0 \\ \frac{1}{\omega\varepsilon_0\varepsilon_t^{(1)}} (P_1e^{ik_z^{(1)}z} + P_2e^{-ik_z^{(1)}z}), & -d_1 < z \leq 0 \\ \frac{1}{\omega\varepsilon_0\varepsilon_t^{(2)}} (P_3e^{ik_z^{(2)}z} + P_4e^{-ik_z^{(2)}z}), & -d < z \leq -d_1 \\ \frac{-iQ_s}{\omega\varepsilon_0\varepsilon_s} (P_3e^{-ik_z^{(2)}d} + P_4e^{ik_z^{(2)}d})e^{-Q_s(z+d)}, & z \leq -d \end{cases} \quad (S8)$$

Where $Q_{a,s} = \sqrt{-\varepsilon_{a,s}k_0^2 + q^2}$, and $k_z^{(1,2)} = \sqrt{\varepsilon_t^{(1,2)}k_0^2 - (\varepsilon_t^{(1,2)}/\varepsilon_z^{(1,2)})q^2}$.

Considering that the tangential components of \vec{E} and \vec{H} are continuous at the interface,

$$\begin{cases} E_x^{(0)} = E_x^{(1)}, & H_y^{(0)} = H_y^{(1)}, & z = 0 \\ E_x^{(1)} = E_x^{(2)}, & H_y^{(2)} = H_y^{(3)}, & z = -d_1 \\ E_x^{(2)} = E_x^{(3)}, & H_y^{(2)} = H_y^{(3)}, & z = -d \end{cases} \quad (S9)$$

Thus the coefficients (P_1-P_4) can satisfy the $M \cdot (P_1, P_2, P_3, P_4)^T = 0$,

$$M = \begin{pmatrix} \frac{iQ_a}{\varepsilon_a} - \frac{k_z^{(1)}}{\varepsilon_t^{(1)}} & \frac{iQ_a}{\varepsilon_a} + \frac{k_z^{(1)}}{\varepsilon_t^{(1)}} & 0 & 0 \\ \frac{k_z^{(1)}}{\varepsilon_t^{(1)}} e^{-ik_z^{(1)}d_1} & -\frac{k_z^{(1)}}{\varepsilon_t^{(1)}} e^{ik_z^{(1)}d_1} & -\frac{k_z^{(2)}}{\varepsilon_t^{(2)}} e^{-ik_z^{(2)}d_1} & \frac{k_z^{(2)}}{\varepsilon_t^{(2)}} e^{ik_z^{(2)}d_1} \\ e^{-ik_z^{(1)}d_1} & e^{ik_z^{(1)}d_1} & -e^{-ik_z^{(2)}d_1} & e^{ik_z^{(2)}d_1} \\ 0 & 0 & \left(\frac{iQ_s}{\varepsilon_s} + \frac{k_z^{(2)}}{\varepsilon_t^{(2)}}\right) e^{-ik_z^{(2)}d} & \left(\frac{iQ_s}{\varepsilon_s} - \frac{k_z^{(2)}}{\varepsilon_t^{(2)}}\right) e^{ik_z^{(2)}d} \end{pmatrix}$$

When the determinant $\det\{M\} = 0$, it is guaranteed that there are non-zero solutions for the amplitude coefficients (P_1-P_4) . Specifically, the dispersion relation can be stated as,

$$e^{2n_1} = -\frac{[(n_2-n_3)(n_3-n_4)(n_4+n_5)]e^{ik_z^{(1)}d_1} + [(n_2+n_3)(n_3+n_4)(n_4+n_5)]e^{-ik_z^{(1)}d_1}}{[(n_2-n_3)(n_3+n_4)(n_4-n_5)]e^{ik_z^{(1)}d_1} + [(n_2+n_3)(n_3-n_4)(n_4-n_5)]e^{-ik_z^{(1)}d_1}} \quad (S10)$$

Where $n_1 = -ik_z^{(2)}d_1 + ik_z^{(2)}d$, $n_2 = \frac{iQ_a}{\varepsilon_a}$, $n_3 = \frac{k_z^{(1)}}{\varepsilon_t^{(1)}}$, $n_4 = \frac{k_z^{(2)}}{\varepsilon_t^{(2)}}$, and $n_5 = \frac{iQ_s}{\varepsilon_s}$.

Therefore, the IFC of polaritons in hBN/MoO₃ can be obtained by solving equation (S10) with equation (S4). In addition, previous works, which are cited in the revised manuscript, demonstrated the derivative process of the IFC of phonon polaritons in MoO₃-based bilayer heterostructure.”

4. Spoof surface plasmon polaritons is an “artificial” surface plasmon polaritons, which

are usually realized through some artificial microstructure, such as drilling a subwavelength square hole array in a metallic block. Therefore, in the manuscript, it could be better to change “spoof plasmon” as “surface plasmon”.

Reply: We thank the reviewer for the kind reminder. Following the reviewer’s suggestion, we have changed “spoof plasmon” to “surface plasmon” in the revised manuscript.

Changes made in the revised manuscript:

We thank the reviewer for this suggestion, in response to which we made the following change in Lines 126-128 on page 5 of the revised manuscript, as marked in red:

“In the s-SNOM measurement, an obliquely incident infrared light forming an angle of approximately 38° with the sample surface is used to excite the surface plasmons on the Ag nanowire for visualizing the wavefront of the polaritonic reversed CR.”

5. Please add the scale for the IFCs of Fig.3 (c) and Fig.4 (b).

Reply: We thank the reviewer for the helpful reminder. As suggested, we have added the scales for Fig. 3(c) and Fig. 4(b). Please refer to the updated figures in the revised manuscript:

Fig. 3. Asymmetric reversed CR in MoO₃.

Fig. 4. Reversed CR in hBN/MoO₃ heterostructure.

6. The authors said that “Since the plasmon has a positive dispersion while the phonon polaritons supported by MoO₃ have a negative dispersion, the wave vectors (k_e , k_{ph}) change in opposite directions when a dielectric material is inserted between the metal

and MoO₃.” If there are some references, please cite, if no, please prove.

Reply: We thank the reviewer for this comment. We have modified this part in the revised manuscript while also adding relevant references (*Appl. Phys. Lett.* 2022, 120, 113101; *Adv. Optical Mater.* 2022, 10, 2102057). Polaritonic materials with positive group velocity dispersion (i.e., $\partial\omega/\partial k > 0$) and negative group velocity dispersion (i.e., $\partial\omega/\partial k < 0$) can exhibit opposite trends in their dispersion curves when the surrounding dielectric environment changes. For example, when a layer of medium with a refractive index greater than 1 is added around a material with positive group velocity dispersion, its dispersion curve will shift towards higher wavevectors. In contrast, the same layer added around a material with negative group velocity dispersion will cause its dispersion curve to shift towards lower wavevectors (see Figure R9). Similarly, inserting a dielectric material such as hBN can reduce wavevector mismatch between the plasmon (k_e) and phonon polariton (k_{ph}), resulting in higher radiation efficiency of the polaritonic reversed CR.

Figure R9. Variation of the polariton dispersion with different types of group velocities ((a), positive group velocity, $\partial\omega/\partial q > 0$; (b), negative group velocity, $\partial\omega/\partial q < 0$) when a dielectric material is introduced.

Changes made in the revised manuscript:

We thank the reviewer for this question, in response to which we made the following change in Lines 189-202 on Page 7 of the revised manuscript, as marked in red:

“To further increase the radiation angle (θ_k) and improve the radiation efficiency ($\eta \approx |\mathbf{k}_e|/(|\mathbf{k}_{ph}| - |\mathbf{k}_e|)$) of the polaritonic reversed CR, one can increase the

plasmonic wave vector (k_e) or reduce the phonon polaritonic wave vector (k_{ph}). A higher CR efficiency can be achieved by increasing the coupling between plasmons and phonon polaritons (i.e., decreasing the momentum mismatch between the large k_{ph} and low k_e), which makes it easier for plasmons (source) to transfer energy and momentum to phonon polaritons (the excited reversed CR). The increase of k_{ph} and decrease of k_e can be realized by increasing the environmental dielectric constant. This is because the plasmon has a positive dispersion while the phonon polaritons supported by MoO₃ have a negative dispersion. Therefore, an effective way to improve the performance of the reversed CR is to construct a heterojunction system by inserting a dielectric film with a high-refractive index between MoO₃ and metal nanowires. Here, we specifically design hBN/MoO₃ heterostructures (Fig. 4a) for this purpose mainly because: (1) It has a high refractive index and low optical dielectric loss in the frequency range of approximately 958-1010 cm⁻¹; (2) It has an atomically flat surface to protect MoO₃ and thus reduce the defect losses on its surface.”

We now cite the noted previous works *Appl. Phys. Lett.* 2022, 120, 113101; *Adv. Optical Mater.* 2022, 10, 2102057) as the new Refs. 33 and 34:

“33 Shen, J. et al. Hyperbolic phonon polaritons with positive and negative phase velocities in suspended α -MoO₃. *Appl. Phys. Lett.* 120, (2022).

34 Zheng, Z. et al. Tunable Hyperbolic Phonon Polaritons in a Suspended van der Waals α -MoO₃ with Gradient Gaps. *Adv. Opt. Mater.* 10, 2102057, (2022).”

7. There are some errors in this manuscript. For example, $\epsilon_{pz} \cos 2\beta + \epsilon_{py} \sin 2\beta$ should be changed as $\epsilon_{px} \cos 2\beta + \epsilon_{py} \sin 2\beta$. Please check it carefully.

Reply: We thank the reviewer for pointing this typo out. We have corrected it as:

$$\varphi = i\sqrt{\epsilon_z / (\epsilon_x \cos^2 \beta + \epsilon_y \sin^2 \beta)}, \quad (S3)$$

We have also made careful and substantial revisions, which are marked in red in the revised manuscript and supplementary information.

Changes made in the revised manuscript:

We have followed the reviewer's suggestion, in response to which we have made the following change in Line 55 on Page 2 of the revised supplementary information, as marked in red:

$$\varphi = i\sqrt{\varepsilon_z/(\varepsilon_x \cos^2 \beta + \varepsilon_y \sin^2 \beta)}, \quad (\text{S3})$$

References

- 1 Ma, W. *et al.* In-plane anisotropic and ultra-low-loss polaritons in a natural van der Waals crystal. *Nature* **562**, 557-562, (2018).
- 2 Zhao, Y. *et al.* Ultralow-Loss Phonon Polaritons in the Isotope-Enriched α -MoO₃. *Nano Lett.* **22**, 10208-10215, (2022).
- 3 Ni, G. *et al.* Long-Lived Phonon Polaritons in Hyperbolic Materials. *Nano Lett.* **21**, 5767-5773, (2021).
- 4 Hu, D., Luo, C., Kang, L., Liu, M. & Dai, Q. Few-layer hexagonal boron nitride as a shield of brittle materials for cryogenic s-SNOM exploration of phonon polaritons. *Appl. Phys. Lett.* **120**, 161101, (2022).
- 5 Shen, J. *et al.* Hyperbolic phonon polaritons with positive and negative phase velocities in suspended α -MoO₃. *Applied Physics Letters* **120**, 113101, (2022).
- 6 Zheng, Z. *et al.* Tunable Hyperbolic Phonon Polaritons in a Suspended van der Waals α -MoO₃ with Gradient Gaps. *Advanced Optical Materials* **10**, 2102057, (2022).
- 7 Menabde, S. G. *et al.* Low-Loss Anisotropic Image Polaritons in van der Waals Crystal α -MoO₃. *Advanced Optical Materials* **10**, 2201492, (2022).
- 8 Hu, H. *et al.* Active control of micrometer plasmon propagation in suspended graphene. *Nat. Commun.* **13**, 1465, (2022).
- 9 Woessner, A. *et al.* Highly confined low-loss plasmons in graphene-boron nitride heterostructures. *Nat. Mater.* **14**, 421-425, (2015).
- 10 Fei, Z. *et al.* Gate-tuning of graphene plasmons revealed by infrared nano-imaging. *Nature* **487**, 82-85, (2012).
- 11 Dai, S. *et al.* Tunable Phonon Polaritons in Atomically Thin van der Waals Crystals of Boron Nitride. *Science* **343**, 1125-1129, (2014).
- 12 Giles, A. J. *et al.* Ultralow-loss polaritons in isotopically pure boron nitride. *Nat. Mater.* **17**, 134-139, (2018).
- 13 Fali, A. *et al.* Refractive Index-Based Control of Hyperbolic Phonon-Polariton Propagation. *Nano Lett.* **19**, 7725-7734, (2019).
- 14 Dai, S. *et al.* Hyperbolic Phonon Polaritons in Suspended Hexagonal Boron Nitride. *Nano Lett.* **19**, 1009-1014, (2019).
- 15 Ma, W. *et al.* In-plane anisotropic and ultra-low-loss polaritons in a natural van der Waals crystal. *Nature* **562**, 557-562, (2018).
- 16 Zheng, Z. *et al.* A mid-infrared biaxial hyperbolic van der Waals crystal. *Science Advances* **5**, eaav8690.

Reviewers' Comments:

Reviewer #1:

Remarks to the Author:

The reviewer carefully checked the replies and found that the revised manuscript was improved.

Here, more comments are listed below:

- 1.The authors should change the title. This is because the authors agree that the so called reversed CR is not REAL reversed CR. It is an analog of reversed CR at optical band.
- 2.In the replies, the authors state that in recent years, we only focus on the near-field study of the polaritonic reversed CR in MoO₃, which has the potential for nanoscale on-chip light sources. Please give more explanations why it has the potential for nanoscale on-chip light sources. What is its advantages and disadvantages relative to the existing counterparts? In addition, please compare the nanoscale on-chip light source presented here with that really using free electrons. And then a detailed application scenario should be given.
- 3.If one wants to change the operating frequency, how to do it easily?

Reviewer #2:

Remarks to the Author:

I am pleased with the author's response. Based on the revised manuscript and the response, the paper can be accepted for publication on Nature Communications. It is an interesting topic with promising applications.

Our point-by-point response to the Reviewers:

We thank all referees and the editor for their careful reviewing of this work and the recommendation for the publication of this work. Please, find our point-by-point reply below.

Reviewer #1 (Comments for the Author):

The reviewer carefully checked the replies and found that the revised manuscript was improved.

Reply: We would like to express our gratitude to the reviewer for providing constructive comments on our manuscript, which has helped us improve the manuscript.

Here, more comments are listed below:

1. The authors should change the title. This is because the authors agree that the so called reversed CR is not REAL reversed CR. It is an analog of reversed CR at optical band.

Reply: Thank you for the reviewer's suggestion. We have changed the title from “*Mid-infrared polaritonic reversed Cherenkov radiation in natural anisotropic crystals*” to “*Mid-infrared analogue polaritonic reversed Cherenkov radiation in natural anisotropic crystals*”.

2. In the replies, the authors state that in recent years, we only focus on the near-field study of the polaritonic reversed CR in MoO₃, which has the potential for nanoscale on-chip light sources.
 - (1) Please give more explanations why it has the potential for nanoscale on-chip light sources.
 - (2) What is its advantages and disadvantages relative to the existing counterparts? In addition, please compare the nanoscale on-chip light source presented here with that really using free electrons. And then a detailed application scenario should be given.

Reply: We thank the reviewer for asking this intriguing question.

- (1) The polaritonic reversed CR in MoO₃ has the potential for nanoscale on-chip light sources due to its unique properties, including (I) highly confined in-plane nanoscale radiative light field distribution, (II) easily integrable atomic-level flat van der Waals heterostructure, and (III) the ability to combine with existing Si-based semiconductor processes to obtain compact, low-cost, and efficient light sources that can be directly integrated into microelectronic circuits.
- (2) Compared to conventional counterparts, we have the following additional advantages:

(a) Radiation light sources that are flexible in design and have strong compatibility. The use of negative group velocity dispersion in hyperbolic natural materials achieves reversed CR in different frequency bands, greatly simplifying the design and reducing energy consumption.

(b) Nanoscale radiation sources in-plane integration. Conventional electron radiation sources are usually three-dimensional and emitted into out-of-plane space. In contrast, polaritonic reversed CR maintains the great wave vector compression ability of phonon polaritons, which can be highly confined on the MoO₃ surface to form the on-chip 2D CR. The radiation power is not yet very high and need to be further improved through optimization of structures and materials.

One promising application of polaritonic reversed CR is in the field of on-chip photonics. The highly confined in-plane nanoscale radiative light field distribution of polaritonic reversed CR on MoO₃ surface makes it an attractive candidate for nanoscale light sources that can be directly integrated into optoelectronic circuits (*Nature*, 2021, 597, 187.).

3. If one wants to change the operating frequency, how to do it easily?

Reply: By screening and designing various types of hyperbolic polaritonic materials, it is possible to achieve reversed CRs at different operating frequencies. With the discovery of a wide range of Type I hyperbolic materials, the hyperbolic frequency band can now cover a range from the terahertz to visible frequency ranges, and our experimental strategy can be applied to a large number of them (*Nat. Commun.*, 2017, 8, 320). Furthermore, by constructing vdW heterostructures, it will be possible to integrate the working bands of reversed CRs in different materials (*Nature*, 2021, 597, 187; *Nat. Rev. Phys.*, 2022, 4, 578).

Reviewer #2 (Comments for the Author):

I am pleased with the author's Reply. Based on the revised manuscript and the Reply, the paper can be accepted for publication on Nature Communications. It is an interesting topic with promising applications.

Reply: We thank the reviewer for the constructive comment on our manuscript. The reviewer's suggestions for our work prompt us to improve the manuscript. We also appreciate the reviewer for recommending our publication in Nature Communications.

Best regards

The authors